Comparative proteome analysis reveals VPS28 regulates milk fat synthesis through ubiquitylation in bovine mammary epithelial cells

Liu Lily 1
Zhang Qin 2 qzhang@sdau.edu.cn
1 College of Life Science, Southwest Forestry University , Kunming , China
2 College of Animal Science and Technology, Shandong Agricultural University , Tai’an , Shandong , China
Li Cong-Jun
Electronic publication date: 2020 Jul 28
Publication date: 2020
Volume: 8
Electronic Location ID: e9542
Received 2020 Apr 13; Accepted 2020 Jun 24
Copyright: © 2020 Liu and Zhang
Copyright year: 2020
Copyright holder: Liu and Zhang
License: This is an open access article distributed under the terms of the Creative Commons Attribution License, which permits unrestricted use, distribution, reproduction and adaptation in any medium and for any purpose provided that it is properly attributed. For attribution, the original author(s), title, publication source (PeerJ) and either DOI or URL of the article must be cited.
License URL: https://creativecommons.org/licenses/by/4.0/

Keywords: VPS28, Milk fat synthesis, iTRAQ, Proteome, Ubiquitylation

Funding: National Natural Science Foundations of China 31902152 National Major Development Program of Transgenic Breeding 2014ZX0800953B National Natural Science Foundations of China 31201772 National Science and Technology Programs of China 2013AA102504 This work was financially supported by the National Natural Science Foundations of China (31902152), the National Major Development Program of Transgenic Breeding (2014ZX0800953B), the National Natural Science Foundations of China (31201772), the National Science and Technology Programs of China (2013AA102504). The funders had no roles in the study design, data collection and analysis, decision to publish or preparation of the manuscript.

==============================
In our previous study, we found that VPS28 (vacuolar protein sorting 28 homolog) could alter ubiquitylation level to regulate milk fat synthesis in bovine primary mammary epithelial cells (BMECs). While the information on the regulation of VPS28 on proteome of milk fat synthesis is less known, we explored its effect on milk fat synthesis using isobaric tags for relative and absolute quantitation assay after knocking down VPS28 in BMECs. A total of 2,773 proteins in three biological replicates with a false discovery rate of less than 1.2% were identified and quantified. Among them, a subset of 203 proteins were screened as significantly down-(111) and up-(92) regulated in VPS28 knockdown BMECs compared with the control groups. According to Gene Ontology analysis, the differentially expressed proteins were enriched in the “proteasome,” “ubiquitylation,” “metabolism of fatty acids,” “phosphorylation,” and “ribosome.” Meanwhile, some changes occurred in the morphology of BMECs and an accumulation of TG (triglyceride) and dysfunction of proteasome were identified, and a series of genes associated with milk fat synthesis, ubiquitylation and proteasome pathways were analyzed by quantitative real-time PCR. The results of this study suggested VPS28 regulated milk fat synthesis was mediated by ubiquitylation; it could be an important new area of study for milk fat synthesis and other milk fat content traits in bovine.

Introduction

VPS28 is a member of the class E VPS proteins, and also is a major component of ESCRT І (endosomal sorting complexes required for transport І). ESCRT-0, ESCRT-І, ESCRT-ІІ, ESCRT-III and some auxiliary components constitute ESCRTs, play crucial roles in concentration and sorting of ubiquitinated proteins of the multivesicular body for incorporation into intralumenal vesicles (Teo et al., 2006; Pineda-Molina et al., 2006; Saksena et al., 2007). The importance of ESCRTs was demonstrated by Raymond et al. (1992), who showed that disruption of ESCRTs resulted in an accumulation of membrane proteins and no longer degraded in the vacule. Recent studies showed that ESCRTs play a critical role in degradation of ubiquitinated proteins through lysosome and proteasom (Ciechanover, 1994; Katzmann, Babst & Emr, 2001). Particularly, VPS28 is localized to ubiquitin-rich endosomes during ligand-induced receptors internalization and contributes directly to receptor trafficking (Bishop, Horman & Woodman, 2002).

In previous studies, we found that a SNP in the 5′ UTR region of VPS28 showed a very strong association with milk fat percentage, and that its expression level significantly affected milk fat synthesis in Chinese Holstein (Liu et al., 2018; Liu & Zhang, 2019; Jiang et al., 2010, 2014). Based on the structural feature and function of ESCRTs, we believed that VPS28 could regulate milk fat synthesis through engaging ESCRTs complexes to affect the ubiquitin-mediated degradation of proteins, which has been proved in our previous study (Liu et al., 2018; Lily Liu, 2019). However, the molecular mechanisms of VPS28 response to milk fat synthesis still remain unclear. Thus, in present study, to better understand the mechanisms, we used isobaric tags for relative and absolute quantitation (iTRAQ) technology that allows quantitative comparisons of protein abundance to much greater insight into the regulation of VPS28 on milk fat synthesis in Chinese Holstein. After performing an RNAi experiment in bovine primary mammary epithelial cells (BMECs), we compared the knockdown BMECs groups with the control groups to identify differentially expressed proteins by iTRAQ. Changes in the expression patterns of the proteins could provide a basis for clarifying the molecular mechanisms for VPS28 regulating milk fat synthesis in Chinese Holstein, which may be a forward step for milk fat synthesis regulation, In addition, this study can provide a reference for elucidating the molecular mechanisms of milk fat traits.

Materials and Methods

Animals

The procedures of collecting BMECs from the mammary tissues of Chinese Holstein cows according to the Animal Welfare Committee of Shandong Agricultural University (Permit Number is SDAUA-2018-022).

Cell culture

Cell culture experiments were performed using primary BMECs. Chemicals were purchased from Life Technologies (Carlsbad, CA, USA) unless noted otherwise. Primary BMECs were kept in our laboratory. BMECs were plated in serum-containing medium DMEM-F12 supplemented with 10 kU/mL penicillin, 10 mg/mL streptomycin, 10% fetal bovine serum and 1% ITS-G (1 mg/mL Insulin, 0.55 mg/mL Transferrin, 0.67 mg/L Selenium Solution). All cells were cultured on plastic cell culture plates at 37 °C in a humidified atmosphere containing 5% CO2.

Knockdown of VPS28 via RNAi in BMECs

Stealth RNAi™ siRNAs targeting the bovine VPS28 gene open reading frame were designed and synthesized by GenePharma Corporation (Shanghai, China). One day prior to transfection, BMECs were seeded without antibiotics. When cells reached 80% confluence, VPS28-siRNAs (GUCCAGGGCUCAGAAAUCATT and GACGUGGUCUCGCUCUUUATT) as tandem constructs, were transfected in to BMECs using X-treme GENE siRNA Transfection Reagent (Roche, Penzberg, Germany) at a 1:10 molar ratio. Cells were harvested at 72 h after transfection for mRNA analysis via real-time quantitative PCR (RT-qPCR).

Sample preparation

Six BMECs samples from two groups (control and VPS28 knockdown) were incubated in lysis buffer (7M urea, 2M thiourea and 0.1% CHAPS) for 30 min on ice and sonicated (80W, ultrasonic 0.2 s, intermittent 2 s, a total 60 s) on ice. Cells debris was pelleted by centrifugation at 15,000×g for 20 min at 4 °C. The supernatants were collected and stored at −80 °C. The protein concentration was determined using Bradford assay (Sigma-Aldrich, St. Louis, MO, USA).

Protein digestion and iTRAQ labeling

Protein digestion was performed using the filter aided sample preparation method. Each protein extract (200 µg) was mixed with 4 µL reducing reagent (AB Sciex, Redwood City, CA, USA) for 1 h at 60 °C and 2 µL cysteine-blocking reagent for 10 min at room temperature, the alkylated protein solution was added to 10 K ultrafiltration tube and discarded the filtrate after centrifuging at 12,000×g for 20 min. Then 100 µL dissolution buffers were added to the filtered unit and the solution was centrifuged again at 12,000×g for 20 min and repeated three times. After incubating overnight, the units were transferred to new collection tubes, and then adding 4 µg trypsin (protein to enzyme ratio 50:1 w/w) and mixed them at 37 °C for overnight. The units were centrifuged at 12,000×g for 20 min discarded the filtrate, then added 50 µL dissolution buffer 5 and centrifuged 12,000×g for 20 min incubated at room temperature. Finally, the extracted peptides were collected from bottom.

The iTRAQ labeling was performed according to the manufacturer’s protocol (AB Sciex, Redwood City, CA, USA). After trypsin digestion, the peptides were transferred to vials containing individual iTRAQ regents by incubation at room temperature for 2 h, which was thawed and reconstituted in 150 μL isopropanol per one unit. The three knock-down VPS28 groups were labeled with iTRAQ 115, 116 and 117; the three WT groups were labeled with iTRAQ 118, 119 and 121, respectively.

Peptide fractionation with strong cation exchange chromatography

The iTRAQ labeled peptides were fractionated by SCX using RIGOL L-3000 HPLC system (RIGOL, Beijing, China). The dried peptide was dissolved with 100 μL buffer A (98% ddH2O, 2% acetonitrile) and the solution was centrifuged at 14,000×g for 20 min, the supernatants were collected. The peptides were eluted at a flow rate of 0.7 mL/min with a buffer B (98% acetonitrile, 2% H2O) gradient of 5% at 0–5 min, 8% at 5–35 min, 18% at 35–62 min, 32% at 62–64 min, 95% at 64–68 min, 5% at 72 min. The elution was monitored by absorbance at 214 nm.

Quantitative analysis of proteins by iTRAQ LC-MS/MS

Each collected component of the processed SCX fractions was redissolved with 20 µL 2% methanol and 0.1% formic acid, and the solution was centrifuged at 12,000×g for 10 min, the supernatants were collected. 10 μL solution was trapped on a precolumn (100 μm × 2 cm) and then eluted on an analytical column (75 μm × 12 cm) for separation. The precolumn was packed with Acclaim PepMap-C18 5 μm and analytical column was packed with EASY-Spray-C18 3 µm. The peptides were separated over 90 min and eluted at a flow rate of 350 nL/min. The MS analysis was performed using an Applied Biosystems Q-Exactive mass spectrometer.

The BMECs iTRAQ identification and quantification analysis were obtained using Proteome Discoverer1.3 (Thermo, Waltham, MA, USA). Proteome Discoverer1.3 was set up to search the NCBI Bos taurus major database assuming the digestion enzyme trypsin. The differential expressed proteins were accepted if they have been identified with greater than 95% confidence in all iTRAQ preparations, and have ≥1.2 or ≤0.83 fold changes (iTRAQ ratios (VPS28 knockdown)-115+116+117: (control)-118+119+121) in addition to P ≤ 0.05. Gene ontology (GO) was used to annotate the proteins under the biological progress (BP), molecular function (MF) and cellular components (CC) GO categories (DAVID, https://david.ncifcrf.gov/) in the BMECs.

Microscopy analysis

The control and VPS28 knockdown BMECs were collected and fixed with 2.5% glutaraldehyde at 4 °C for overnight, and washed by PBS (pH 7.0, 0.1M) for three times. And then the BMECs was fixed with 1% osmium tetroxide for 1–2 h, washed by sodium cacodylate buffer, and then dehydrated with gradient alcohol until complete, finally embedded in Epon 812. The fixed BMECs were cut into 1-um-thick sections and stained with uranyl acetate and lead citrate. The ultrathin sections were examined under JEM-1400 electron microscope (JEOL, Tokyo, Japan).

Measurement of cellular TG content and proteasome activities

The control and VPS28 knockdown BMECs were collected and broken by ultrasonication. The total lipids were extracted using the TG assay Kit (Nanjing Jiancheng Bioengineering Institute, Jiangsu, China) and monitored with Infinite M200 Reader (Tecan, Männedorf, Switzerland) according to the manufacturer’s instructions.

The proteasome activities (Chymotrypsin-Like, Caspase-Like and Trypsin-Like) were measured using the Proteasome-Glo™ Cell-Based Assays (Promega, Mannheim, Germany) according to the manufacturer’s instructions, and the fluorescence intensity was monitored with Infinite M200 Reader (Tecan, Männedorf, Switzerland).

Real-time quantitative PCR analysis

The primers of selected genes for RT-qPCR were designed with Primer 5.0 and synthesized by The Beijing Genomics Institute Co., Ltd. The glyceraldehyde-3-phosphate dehydrogenase gene was used as the control. The primer sequences are listed in Table 1. RNA extraction, cDNA synthesis and RT-qPCR were performed according to the manufacturer’s instructions, and were repeated three times. The relative expression of genes was computed using the 2−ΔΔCt method.

Table 1 Differentially expressed proteins following VPS28 knockdown in BMEC.

Genes	Primer sequences (5′→3′)	Relative expression	
GAPDH	AGATGGTGAAGGTCGGAGTG
CGTTCTCTGCCTTGACTGTG	/	
VPS28	GGAAACAAGCCGGAGCTGTA
CTGGATCTCGTCCATGGCTC	0.22	
CD36	GACGGATGTACAGCGGTGAT
GAAAAAGTGCAAGGCCACCA	16.00	
ACACA	AGTGTTCTGATCAGGTCTTCTTGT
GGGAGGCAAAAACCTCCAGA	0.67	
FASN	AGGCGTGCGTGACACTT
AATACAGTTGGCCGTCACCA	6.85	
SCD	TCCTGATCATTGGCAACACCA
CCAACCCACGTGAGAGAAGAA	1.48	
DGAT1	TACCCCGACAACCTGACCTA
GGGAAGTTGAGCTCGTAGCA	2.06	
ADFP	GCGTCTGCTGGCTGATTTC
AGCCGAGGAGACCAGATCATA	2.95	
PSMG1	GGGAAGAAGTCGGTTGTGCT
AAAAAGCCTCTGTGGGGGAC	2.87	
UBE2L	CTGGCACAGTATATGAAGACCTGA
GGTAGCAGGGTGTGAGGAAC	1.28	
RPS29	TGTTTCCGCCAGTATGCGAA
GCTGGATGAGCCATCTAAGGAA	2.13	
ISG15	CCATCCTGGTGAGGAACGAC
GTCTGCTTGTACACGCTCCT	19.02	

Statistical Analysis

R-package (R v3.02) was conducted to evaluated changes between VPS28 knockdown BMECs groups and the control groups. And differences were declared significant at P ≤ 0.05.

Results

VPS28 knockdown alters expression of multiple proteins in BMEC

VPS28 expression in BMEC were down-regulated by 78% with tandem constructs (as shown in Fig. 1A), and then, to obtain a whole picture of the proteomic changes in VPS28 knockdown BMEC, we conducted iTRAQ experiment in combination with LC-ESI-MS/MS analysis to investigate differentially expressed proteins in VPS28 knockdown BMECs groups (labeled iTRAQ-115, 116 and 117) and the control groups (labeled iTRAQ-118, 119 and 121). At a false discovery rate of 1.2%, a total of 2,773 proteins were identified from 14,031 peptides. The peptides of all proteins are provided in Table S1.

Figure 1 Effects of VPS28 knockdown on BMECs.

(A) The mRNA expression of VPS28 was decreased by tandem siRNAs constructs. (B) and (C) Electron micrographs of BMECs. (D) The TG content was significantly increased in VPS28 knockdown BMEC. Data are averages of three replicates. The error or bars denote SEM. *Indicates the difference is significant (P ≤ 0.05).

To further understand the differentially expressed proteins after knocking down VPS28 in BMECs and basing on standard of the differentially expressed proteins, a total of 203 distinct proteins were identified by iTRAQ analysis in VPS28 knockdown BMECs (Detailed gene information and fold-change following VPS28 knockdown were provided in Table 2). A total of 92 proteins were significantly up-regulated (≥1.2-fold) while 111 proteins were significantly down-regulated (≤0.83-fold) when compared with the control BMECs.

Table 2 Gene ontology analysis of differentally expressed proteins in VPS28 knockdown BMECs.

Accession	Description	Ratio	
300796460	Pescadillo homolog	0.79	
333440457	Immortalization up-regulated protein	1.38	
528937065	PREDICTED: fragile X mental retardation syndrome-related protein 1 isoform X6	0.83	
741896620	PREDICTED: bifunctional coenzyme A synthase isoform X2	1.25	
741972182	PREDICTED: serpin B8 isoform X1	1.33	
359069079	PREDICTED: apoptotic chromatin condensation inducer in the nucleus isoform X5	0.74	
528978576	PREDICTED: lysosomal acid phosphatase isoform X3	0.8	
77736117	Actin, alpha cardiac muscle 1	1.33	
741976470	PREDICTED: actin filament-associated protein 1-like 2 isoform X5	0.82	
156120791	A-kinase anchor protein 8	0.62	
155371939	Putative N-acetylglucosamine-6-phosphate deacetylase	1.23	
741911242	PREDICTED: AP-2 complex subunit sigma-like	0.82	
115496866	AP-3 complex subunit beta-1	1.22	
75832056	Apolipoprotein A-I preproprotein	0.46	
114052298	Apolipoprotein A-II precursor	0.55	
741944057	PREDICTED: apolipoprotein B-100 isoform X3	0.71	
27806739	Apolipoprotein E precursor	0.34	
51491835	Ovarian and testicular apolipoprotein N precursor	0.61	
528973530	PREDICTED: ADP-ribosylation factor GTPase-activating protein 1 isoform X3	0.8	
300798482	Rho GTPase-activating protein 35	1.36	
329664977	AT-rich interactive domain-containing protein 1A	0.83	
529009701	PREDICTED: acid ceramidase isoform X1	1.62	
329663402	ATPase family AAA domain-containing protein 1	0.76	
741980112	PREDICTED: atlastin-3 isoform X1	0.83	
60101829	ATP synthase subunit 8 (mitochondrion)	1.24	
28603752	ATP synthase subunit e, mitochondrial	0.63	
116004323	Ataxin-10	0.83	
41386683	Beta-2-microglobulin precursor	0.82	
84000125	B-cell receptor-associated protein 29	1.34	
27806229	2-oxoisovalerate dehydrogenase subunit alpha, mitochondrial precursor	0.83	
741929024	PREDICTED: uncharacterized protein C4orf3 homolog isoform X1	1.35	
741945468	PREDICTED: calcium-binding protein 39-like isoform X1	0.79	
45439308	CD63 antigen	1.36	
78042548	CD81 antigen	1.25	
741967799	PREDICTED: LOW QUALITY PROTEIN: serine/threonine-protein kinase MRCK beta isoform X2	1.37	
529002260	PREDICTED: CUB domain-containing protein 1	0.8	
77735577	CCR4-NOT transcription complex subunit 7	0.77	
741922497	PREDICTED: collagen alpha-3(VI) chain isoform X7	0.81	
114052042	COMM domain-containing protein 1	1.23	
741945876	PREDICTED: COMM domain-containing protein 6 isoform X2	0.68	
528966533	PREDICTED: COP9 signalosome complex subunit 2 isoform X1	0.82	
149642865	COP9 signalosome complex subunit 3	0.74	
330688478	Crooked neck-like protein 1	1.6	
262073106	Cathepsin D precursor	1.23	
118151448	CUGBP Elav-like family member 2	1.25	
741917150	PREDICTED: cytochrome P450 20A1 isoform X1	1.27	
164420721	Dynactin subunit 5	1.29	
528937089	PREDICTED: DCN1-like protein 1 isoform X1	0.67	
149642575	ATP-dependent RNA helicase DDX24	1.54	
114051872	Density-regulated protein	0.81	
157427916	H/ACA ribonucleoprotein complex subunit 4	1.22	
115497846	deoxyhypusine hydroxylase	1.48	
528989517	PREDICTED: developmentally-regulated GTP-binding protein 1 isoform X1	1.47	
114051994	Dysbindin	1.35	
329663806	Cytoplasmic dynein 1 light intermediate chain 2	1.23	
56710336	Dynein light chain 1, cytoplasmic	0.83	
77735949	3-beta-hydroxysteroid-Delta(8),Delta(7)-isomerase	1.24	
62751595	Translation initiation factor eIF-2B subunit beta	0.71	
300794424	Eukaryotic translation initiation factor 5	0.78	
329664532	Ephrin type-A receptor 2 precursor	0.74	
77735625	Enhancer of rudimentary homolog	0.81	
27806943	Coagulation factor V precursor	0.81	
528957418	PREDICTED: protein FAM114A2 isoform X1	1.29	
329663573	Protein FAM134A	0.75	
359069460	PREDICTED: protein FAM98B	0.8	
29135293	Farnesyl pyrophosphate synthase	0.77	
77736507	Mitochondrial fission 1 protein	1.27	
156718120	Fat storage-inducing transmembrane protein 2	1.23	
27806621	Ferritin heavy chain	0.8	
114051796	Glucosylceramidase precursor	0.81	
84000253	Glutamate--cysteine ligase regulatory subunit	1.21	
114051291	GDP-L-fucose synthase	0.78	
741919465	PREDICTED: lysosomal protein NCU-G1 isoform X2	0.81	
115496402	Glucosamine-6-phosphate isomerase 2	0.83	
297488836	PREDICTED: histone H1x	0.82	
116812902	Hemoglobin subunit alpha	0.55	
17985949	Hemoglobin subunit beta-1 [Rattus norvegicus]	1.23	
741905547	PREDICTED: host cell factor 1 isoform X9	1.26	
114052627	Hepatocyte growth factor-regulated tyrosine kinase substrate	1.21	
134085671	Histone H1.2	0.55	
155371863	Histone H1.3	0.54	
741971316	PREDICTED: histone H2A type 1-J	1.29	
157785601	Histone H2B	0.82	
115496175	High mobility group protein HMG-I/HMG-Y	0.8	
77736489	Non-histone chromosomal protein HMG-14	0.79	
297477251	PREDICTED: heterogeneous nuclear ribonucleoprotein A0	0.83	
375364520	HCLS1-binding protein 3	1.48	
41386699	Heat shock-related 70 kDa protein 2	1.23	
529014943	PREDICTED: immunoglobulin-binding protein 1 isoform X2	1.23	
27805955	Ubiquitin-like protein ISG15	0.83	
157427772	Involucrin	1.25	
195539527	Keratin 15	1.24	
77736483	Ragulator complex protein LAMTOR1	0.82	
741894288	PREDICTED: galectin-7	1.26	
528952868	PREDICTED: LIM and calponin homology domains-containing protein 1 isoform X5	1.36	
115497506	LIM and cysteine-rich domains protein 1	1.23	
686713724	PREDICTED: LOW QUALITY PROTEIN: collagen alpha-4(VI) chain-like, partial [Pongo abelii]	0.72	
741878073	PREDICTED: N-acylneuraminate cytidylyltransferase	1.96	
741946731	PREDICTED: ankyrin repeat domain-containing protein 26-like isoform X2	0.35	
62460494	Hemoglobin fetal subunit beta	0.56	
84000167	WD repeat-containing protein 61	1.36	
741960002	PREDICTED: protein arginine N-methyltransferase 1 isoform X2	0.76	
297483902	PREDICTED: apolipoprotein R	0.61	
155372051	Tropomyosin alpha-4 chain	0.83	
78369240	U6 snRNA-associated Sm-like protein LSm4	0.7	
122692397	Latexin	0.79	
77735445	Protein mago nashi homolog	1.22	
741939300	PREDICTED: dual specificity mitogen-activated protein kinase kinase 1 isoform X1	1.21	
528995215	PREDICTED: dual specificity mitogen-activated protein kinase kinase 4 isoform X2	1.21	
741898851	PREDICTED: MAP/microtubule affinity-regulating kinase 3 isoform X1, partial	0.83	
528957564	PREDICTED: methionine adenosyltransferase 2 subunit beta isoform X1	0.81	
528966905	PREDICTED: protein max isoform X2	0.76	
741957547	PREDICTED: mediator of RNA polymerase II transcription subunit 15 isoform X3	0.79	
300794942	DNA mismatch repair protein Msh6	0.74	
27806841	interferon-induced GTP-binding protein Mx1	0.73	
528936325	PREDICTED: N-alpha-acetyltransferase 50 isoform X1	1.3	
375065860	NAD kinase 2, mitochondrial	1.29	
300795748	NEDD8-activating enzyme E1 regulatory subunit	0.77	
331284195	Nucleolin	1.38	
78369204	Protein NDRG2	1.24	
28372495	NADH dehydrogenase [ubiquinone] 1 alpha subcomplex subunit 11	1.23	
75812936	NADH dehydrogenase [ubiquinone] 1 beta subcomplex subunit 11, mitochondrial precursor	1.32	
28603776	NADH dehydrogenase [ubiquinone] 1 beta subcomplex subunit 5, mitochondrial precursor	0.72	
528944090	PREDICTED: nexilin isoform X5	0.76	
300794221	Nuclear protein localization protein 4 homolog	0.82	
83035119	Nuclear transport factor 2	0.79	
741958202	PREDICTED: prolyl 3-hydroxylase OGFOD1 isoform X1	0.65	
27807193	Platelet-activating factor acetylhydrolase IB subunit beta	1.27	
75812940	Phosphatidylethanolamine-binding protein 1	1.27	
528913445	PREDICTED: presequence protease, mitochondrial isoform X2	1.21	
329664500	Pyruvate kinase PKM	1.25	
528961976	PREDICTED: pyruvate kinase PKM isoform X1	1.26	
741932605	PREDICTED: perilipin-3 isoform X3	0.82	
116004039	Peptidyl-prolyl cis-trans isomerase C precursor	1.22	
741957590	PREDICTED: protein phosphatase 1F	1.53	
115497768	RelA-associated inhibitor	1.24	
528943961	PREDICTED: cAMP-dependent protein kinase catalytic subunit beta isoform X8	1.22	
741948151	PREDICTED: pre-mRNA-processing factor 6 isoform X1	0.73	
115496548	Proteasome assembly chaperone 1	1.26	
741926509	PREDICTED: prostaglandin E synthase 3 isoform X1	0.83	
157428086	Ras-related protein Rab-8A	0.8	
77736231	Ras-related protein Ral-A	1.2	
56118252	RING finger protein 113A	1.27	
741937627	PREDICTED: ribosome production factor 2 homolog isoform X1	0.77	
27807465	60S ribosomal protein L10	0.79	
62751646	60S ribosomal protein L13	0.7	
116004215	60S ribosomal protein L13a	0.81	
118150852	60S ribosomal protein L15	0.7	
62751887	60S ribosomal protein L26	0.77	
77404275	60S ribosomal protein L27	0.79	
77735585	60S ribosomal protein L36a	0.8	
62460480	60S ribosomal protein L4	0.74	
114053031	39S ribosomal protein L48, mitochondrial precursor	0.81	
72534798	60S ribosomal protein L6	0.72	
62460552	60S ribosomal protein L7	0.77	
77736197	60S ribosomal protein L8	0.78	
164420694	60S ribosomal protein L9	1.21	
70778762	60S acidic ribosomal protein P1	1.2	
66792924	40S ribosomal protein S11	0.76	
77735975	28S ribosomal protein S26, mitochondrial precursor	1.23	
528994013	PREDICTED: 28S ribosomal protein S23, mitochondrial isoform X3	1.23	
27807381	40S ribosomal protein S29	1.23	
70778956	40S ribosomal protein S8	0.79	
155372029	40S ribosomal protein S9	0.69	
741947465	PREDICTED: ribosome-binding protein 1 isoform X2	1.31	
300798287	Sec1 family domain-containing protein 1	0.83	
115497454	Protein SEC13 homolog	1.28	
300794266	SEC23-interacting protein	1.3	
741978352	PREDICTED: protein transport protein Sec24C isoform X2	1.31	
115497008	Protein transport protein Sec61 subunit beta	0.82	
70778796	Splicing factor 3B subunit 5	1.3	
77736509	S-phase kinase-associated protein 1	1.35	
82617542	Monocarboxylate transporter 1	1.24	
288557348	SWI/SNF complex subunit SMARCC2	0.8	
115496404	U1 small nuclear ribonucleoprotein C	0.78	
329664862	S1 RNA-binding domain-containing protein 1	0.73	
741921253	PREDICTED: serine/arginine-rich splicing factor 11 isoform X4	1.49	
329664840	synaptopodin	0.82	
84000143	T-complex protein 1 subunit alpha	1.21	
114051768	Tudor domain-containing protein 3	1.36	
300797062	Tudor domain-containing protein 6	1.52	
529000498	PREDICTED: THUMP domain-containing protein 3 isoform X1	1.3	
114326224	Tight junction protein ZO-3	1.25	
300794719	E3 ubiquitin-protein ligase TRIP12	0.78	
27806789	Transthyretin precursor	1.29	
529005013	PREDICTED: thioredoxin-like protein 1 isoform X2	1.21	
83035103	Ubiquitin-conjugating enzyme E2 H	1.24	
528979920	PREDICTED: ubiquitin conjugation factor E4 B isoform X1	1.22	
114050863	Ubiquitin-like domain-containing CTD phosphatase 1	0.72	
529012185	PREDICTED: UBX domain-containing protein 1 isoform X1	0.78	
62751620	Ubiquitin-fold modifier-conjugating enzyme 1	0.82	
529006388	PREDICTED: ubiquitin carboxyl-terminal hydrolase 7 isoform X2	1.37	
115496338	Vesicle-associated membrane protein-associated protein A	1.2	
78369492	Vacuolar protein sorting-associated protein 28 homolog	0.79	
78045497	Vitronectin precursor	0.43	
741916372	PREDICTED: xin actin-binding repeat-containing protein 2 isoform X2	0.72	
126723764	Cap-specific mRNA (nucleoside-2′-O-)-methyltransferase 1	0.8	
78042540	Synaptobrevin homolog YKT6	0.82	
148224064	Transcriptional repressor protein YY1	0.76	
84370039	Zinc finger protein ZPR1	0.77	
528942220	PREDICTED: rho guanine nucleotide exchange factor 2 isoform X5	1.2	
528962021	PREDICTED: geranylgeranyl transferase type-2 subunit alpha isoform X1	0.83	
528979380	PREDICTED: glyoxylate reductase/hydroxypyruvate reductase	1.35	

The DEPs were categorized into 53 clusters (P < 0.05, as shown in Table 3) according to their biological processes (BPs), cellular components (CCs) and molecular functions (MFs). The top 6 GO terms for BPs were enriched in cytoplasmic translation (GO:0002181), translation (GO:0006412), cholesterol homeostasis (GO:0042632), cholesterol efflux (GO:0033344), positive regulation of cholesterol esterification (GO:0010873) and high-density lipoprotein particle assembly (GO:0034380). These biological processes were involved in the lipid metabolism and transportation. The top 5 GO terms for CCs were cytosolic large ribosomal subunit (GO:0022625), extracellular exosome (GO:0070062), focal adhesion (GO:005925), membrane (GO:0016020) and very-low-density lipoprotein particle (GO:0034361). These cellular components were response to the ubiquitin system. The top 5 GO terms for MFs were mainly enriched in structural constituent of ribosome (GO:0003735), RNA binding (GO:0003723), cholesterol transporter activity (GO:0017127), phosphatidylcholine-sterol O-acyltransferase activator activity (GO:0019843). These results showed that the DEPs following VPS28 knockdown were mainly involved in the functions of transport and metabolism of lipid, lipoprotein and lipoprotein receptor binding, and ribosome translation.

Table 3 Primers of the selected genes for qRT-PCR and their relative expression.

GO ID	Term	P-value	
Biological process	
GO:0002181	Cytoplasmic translation	3.60E−07	
GO:0006412	Translation	8.90E−07	
GO:0042632	Cholesterol homeostasis	2.20E−03	
GO:0033344	Cholesterol efflux	2.70E−03	
GO:0010873	Positive regulation of cholesterol esterification	2.80E−03	
GO:0034380	High-density lipoprotein particle assembly	3.70E−03	
GO:0000463	Maturation of LSU-rRNA from tricistronic rRNA transcript (SSU-rRNA, 5.8S rRNA, LSU-rRNA)	4.70E−03	
GO:0043691	Reverse cholesterol transport	7.10E−03	
GO:0033700	Phospholipid efflux	8.40E−03	
GO:0098779	Mitophagy in response to mitochondrial depolarization	1.00E−02	
GO:0042157	Lipoprotein metabolic process	1.30E−02	
GO:0019433	Triglyceride catabolic process	1.50E−02	
GO:0006904	Vesicle docking involved in exocytosis	2.30E−02	
GO:0000027	Ribosomal large subunit assembly	2.30E−02	
GO:0018158	Protein oxidation	2.30E−02	
GO:0006403	RNA localization	2.30E−02	
GO:0010628	Positive regulation of gene expression	2.60E−02	
GO:0001843	Neural tube closure	3.10E−02	
GO:0006695	Cholesterol biosynthetic process	3.20E−02	
GO:0051028	mRNA transport	3.20E−02	
GO:0042921	Glucocorticoid receptor signaling pathway	3.50E−02	
GO:0010903	Negative regulation of very-low-density lipoprotein particle remodeling	3.50E−02	
GO:0006046	N-acetylglucosamine catabolic process	3.50E−02	
GO:1901998	Toxin transport	3.70E−02	
GO:0006888	ER to Golgi vesicle-mediated transport	3.80E−02	
GO:0018206	Peptidyl-methionine modification	4.60E−02	
GO:0042159	Lipoprotein catabolic process	4.60E−02	
GO:0042158	Lipoprotein biosynthetic process	4.60E−02	
Cellular component	
GO:0022625	Cytosolic large ribosomal subunit	4.00E−12	
GO:0070062	Extracellular exosome	2.70E−08	
GO:0005925	Focal adhesion	1.30E−05	
GO:0016020	Membrane	1.70E−04	
GO:0034361	Very-low-density lipoprotein particle	3.60E−04	
GO:0005840	Ribosome	8.30E−04	
GO:0022627	Cytosolic small ribosomal subunit	9.90E−04	
GO:0042627	Chylomicron	3.30E−03	
GO:0072562	Blood microparticle	9.70E−03	
GO:0034363	Intermediate-density lipoprotein particle	3.30E−02	
GO:0005730	Nucleolus	3.80E−02	
GO:0008180	COP9 signalosome	4.60E−02	
GO:0005737	Cytoplasm	4.70E−02	
Molecular function	
GO:0003735	Structural constituent of ribosome	5.20E−10	
GO:0003723	Poly(A) RNA binding	2.90E−09	
GO:0017127	Cholesterol transporter activity	3.30E−04	
GO:0019843	rRNA binding	5.20E−04	
GO:0060228	Phosphatidylcholine-sterol O-acyltransferase activator activity	1.40E−03	
GO:0003729	mRNA binding	6.90E−03	
GO:0005543	Phospholipid binding	1.70E−02	
GO:0003743	Translation initiation factor activity	2.10E−02	
GO:0003723	RNA binding	3.50E−02	
GO:0008035	High-density lipoprotein particle binding	3.50E−02	
GO:0070653	High-density lipoprotein particle receptor binding	3.50E−02	
GO:0015485	Cholesterol binding	4.60E−02	

Effect of VPS28 knockdown on morphology of BMECs

Electron micrographs could observe the morphological changes in BMECs. Compared with the control BMECs groups, the VPS28 knockdown groups showed containing more and strikingly large lipid droplets and many luminal spaces were completely filled with aggregated lipid (as shown in Figs. 1B and 1C). And in parallel, the content of TG was increased by 3.3-fold above the control BMECs groups (Fig. 1D).

The GO analysis demonstrated DEPs enriched in ubiquitylation singaling, and ubiquitylation mediates the degradation of membrane proteins and intracellular proteins, which plays an crucial role in receptor-mediated signaling pathways and quality control of intracellular proteins. And then we examined the proteasome activity (chymotreypsin-like activity, caspase-like activity, trypsin-like activity) after knocking down VPS28 (as shown in Fig. 2). The results showed that VPS28 knockdown could significantly decrease the three activites of proteasome, the relative activities of chymotreypsin-like, caspase-like and trypsin-like are 0.60, 0.64, 0.74, respectively. And we also found the level of ubiquitinated proteins was increased by VPS28 knockdown (the data has published) (Lily Liu, 2019). These results indicated that VPS28 could regulate ubiquitylation-proteasome system.

Figure 2 The proteasome activity was decreased by VPS28 knockdown.

Chymotreypsin-like activity, caspase-like activity and trypsin-like activity are the three activities in proteasome. An asterisk (*) indicates the difference is significant (P ≤ 0.05).

Validation of gene expression by RT-qPCR

To investigate whether the alteration of proteins expression level were the result of transcriptional regulation, we detected mRNA levels of five selected proteins and five genes that were related to metabolism of fatty acids, ubiquitylation and proteasome pathways. The RT-qPCR results showed high qualitative and quantitative concordance (correlation coefficient > 0.95). As shown in Fig. 3, CD36 (cluster of differentiation 36) in fatty acids taken up process, FASN (fatty acid synthase), SCD (stearoyl-CoA desaturas) and DGAT1 (diacylglycerol acyltransferase 1) in fatty acids synthesis pathway, ADFP (adipose differentiation-related protein) in lipid droplet secretion process, were all up-regulated by VPS28 knockdown. ACACA (acetyl-CoA) in de novo fatty acids synthesis pathway was down-regulated in VPS28 knockdown BMECs. PSMG1 (proteasome assembly chaperone 1) in proteasome system, RPS29 (ribosomal protein S29) in ribosome translation pathway, UBE2L (ubiquitin-conjugating enzyme E2L), ISG15 (interferon-stimulatory gene ISG15) in ubiquitylation pathway, were all up-regulated by VPS28 knockdown. The results showed that the mRNA expression levels of genes were generally corresponded with the changes in the morphology of BMECs and proteins expression detected by the iTRAQ approach.

Figure 3 The mRNA expression of selected genes in VPS28 knockdown BMECs.

Discussion

Milk fat synthesis is a complex process. Numerous types of molecular and chemical relationships exist which directly or indirectly could affect protein activity and regulate milk fat synthesis, such as ubiquitylation and protein–protein interaction. Ubiquitylation is an important post-translational modification and it can mediate the intercellular proteins degradation which plays a crucial role in receptor-mediated signaling pathways. VPS28 as a subunit of ESCRTs is crucial for ubiquitin-mediated degradation of proteins, and we found VPS28 could alter the ubiquitylation level to regulate milk fat synthesis in previous studies (Liu et al., 2018; Lily Liu, 2019). However, much less is understood regarding the molecular mechanisms of VPS28 regulating milk fat synthesis through ubiquitylation. In this study, iTRAQ technology were performed to accurately identify the peptides and precisely quantify the iTRAQ labels. Subsequently, cluster and pathways analysis were devoted to obtain consistent results to further elucidate the regulation pathways of VPS28 on the milk fat synthesis.

The ubiquitin system is a protein degradation pathway, dedicates to the ubiquitylation of cellular targets and the subsequent control of numerous cellular functions and plays an important role in metabolism regulation (Hoeller & Dikic, 2009). The deregulation of components of this elaborate network leads to an accumulation of membrane proteins and no longer degraded in the vacule. Numerous studies indicated that, as one subunit of ESCRTs, VPS28 played a crucial role in ubiquitin-mediated degradation of membrane proteins (Ciechanover, 1994; Katzmann, Babst & Emr, 2001) and cytoplasmic proteins (Smith et al., 2008). In this study, BMECs sections showed that the form of fat droplets was affected after knocking down VPS28, and we found an accumulation of ubiquitinated proteins and a dysfunction of proteasome activity in VPS28 knockdown BMECs groups. The proteomic analysis indentified many differentially expressed proteins that were considerably enriched in extracellular exosome (GO:0070062) and membrane (GO:0016020). These GO categories were associated with the ubiquitylation system. These indicated that VPS28 knockdown played a crucial role in ubiquitylation.

In BMECs, fatty acids taken up and de novo fatty acids synthesis are involved in milk fat synthesis. In our previous study, by knocking down VPS28 in BMECs, we found ubiquitinated CD36 level was increased significantly which is the main protein involved in long-chain fatty acids uptake, and the mRNA expression of other milk fat-related genes were also up-regulated. These results indicated the process of long-chain fatty acids taken up was promoted by VPS28 knockdown in BMECs. In parallel, the expression of ADFP was found increased in VPS28 knockdown BEMCs. ADFP as a specific marker for lipid droplet, its expression level is in keeping with with abundance of lipid droplets in cell (Chang & Chan, 2007). The proteomic analysis also indentified many differentially expressed proteins enriched in lipid metabolism. These data confirmed VPS28 knockdown could facilitate milk fat synthesis in BMECs.

The DEPs analysis indicates that VPS28 could regulate milk fat synthesis in two approaches, the one is VPS28 directly regulates milk fat synthesis through ubiquitylation and the other one is VPS28 mediates ubiquitin-proteasome system to regulate milk fat synthesis. To further understand these, we used the key interact proteins and genes to generate the pathway networks, following is the description of the model presented in Fig. 4:

Figure 4 The network of VPS28 knockdown regulates milk fat synthesis in BMECs.

VPS28 knockdown leads an accumulation of ubiquitinated membrane proteins to promote fatty acids taken up to synthesize TG: In this regulation, CD36 appears to be the most important protein, and the other enzymes involved in milk fat synthesis could be increased through allosteric effect. CD36 as a membrane scavenger receptor was identified as a receptor of fatty acid and ubiquitinated CD36 facilitates long-chain fatty acids uptake (Liang et al., 2004; Schrader, Harstad & Matouschek, 2009; Lamb et al., 2010). Following VPS28 knockdown, the accumulation of ubiquitinated CD36 could import more long-chain fatty acids into BMECs, and the imported long-chain fatty acids are combined and transported to endoplasmic reticulum by fatty acid binding proteins. Subsequently, SCD and DGAT1 are induced to utilize fatty acids to synthesize TG. Therefore, VPS28 knockdown could promote long chain fatty acids taken up to synthesize TG.

VPS28 knockdown leads an accumulation of ubiquitinated cytoplasmic proteins to promote de novo biogenesis, activation and channeling of fatty acids to synthesize TG: In this regulation, proteasome plays the most important role. Following VPS28 knockdown, proteasome activity and the expression of ISG15 (interferon-stimulatory gene ISG15) were decreased. ISG15 is an ubiquitin-like protein that mediates the conjugation of different proteins through its ISGylation enzymes UBE2L6 (ubiquitin conjugating enzyme E2 L6) (Haq et al., 2016), and we also found UBE2L6 was down regulated. Previous studies have suggested that down-regulation of ISG15 and UBE2L6 can counteract degradation of triglyceride lipase (Zhou et al., 2015; Kim et al., 2004; Zhao et al., 2004), and ISG15 conjugation regulates exosome secretion (Villarroya-Beltri et al., 2016). The accumulation of ACACA (Acetyl-CoA) (Emery, 1973) and the other allosteric affected enzymes promote the de novo biogenesis, activation and channeling of fatty acids to synthesize TG in BMECs.

Conclusions

In this study, iTRAQ technology was used to demonstrate proteome spectrum changes in the BMECs after knocking down VPS28. It was concluded that VPS28 knockdown promotes milk fat synthesis in BMECs which might be attributed to differentially expressed proteins. The DEPs enriched in GO categories associated with ubiquitylation likely played an important role in the TG synthesis in BMECs. The dysfunctional proteasome, accumulation of TG might explain the regulation of VPS28 on milk fat synthesis was mediated by ubiquitylation. Our results provide a comprehensive dataset of ubiquitylation regulating milk fat synthesis, and also provide a reference for the further study of ubiquitination in dairy breeding.

Supplemental Information

Supplemental Information 1 Raw data of iTRAQ.

Click here for additional data file.

Supplemental Information 2 qPCR raw data.

Click here for additional data file.

Additional Information and Declarations

Competing Interests

Author Contributions

Animal Ethics

Data Availability

The authors declare that they have no competing interests.

Lily Liu performed the experiments, analyzed the data, prepared figures and/or tables, authored or reviewed drafts of the paper, and approved the final draft.

Qin Zhang conceived and designed the experiments, analyzed the data, prepared figures and/or tables, authored or reviewed drafts of the paper, and approved the final draft.

The following information was supplied relating to ethical approvals (i.e., approving body and any reference numbers):

Animal ethics committee of Shandong Agriculture University provided full approval for this research (SDAUA-2018-022).

The following information was supplied regarding data availability:

Raw measurements are available as a Supplemental File.

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
