# Peer review of "Comparative proteome analysis reveals VPS28 regulates milk fat synthesis through ubiquitylation in bovine mammary epithelial cells"

_PeerJ, doi:10.7717/peerj.9542_

## Round 0.1 · original submission · Minor Revisions

Dear Dr. Zhang,
Thank you very much for your submission to PeerJ. Two reviewers have reviewed your manuscript. While both of the reviewers agree that your manuscript has the merit to be published, there are some minor points that need improvement. Please carefully follow the reviewers' comments to revise your manuscript.

Reviewer 1 ·

Basic reporting

Line 30: enrich:enriched.
Line 33: Abbreviations TG should be explained on their first appearance both in abstract and in the main text.
Line 60: to better understanding: to better understand.
Line 65: these pathways: which pathways? Just saw‘milk fat synthesis’ was mentioned before in main body, ‘these pathways’is not clear.
Line 67: ‘it’ is ambiguous here. Authors should name it clearly, such as: this study or our results, etc.
Line 70: Cells culture should be written as Cell culture.
Line 88-89: It’s not clear for ‘foe mRNA’, please correct or clarify it.
Line 131, 140, 156: These lines were not indented as others. Please check and keep consistent through your manuscript according to PeerJ guidelines.
Line 133: ‘Bos Taurus’here should be written as Bos taurus and italic.
Line 134: ‘identified’ is repeated here.
Line 137-138: The abbreviation for molecular function is missed here.
Line 159: ‘the method described previously’, authors didn’t describe them before, alternatively, literature(s) could be provided.
Line 216: qRT-PCR in heading line has different format from others (RT-qPCR) in Methods, Table 3 and main body (Line 220). Please check all similar words through your manuscript completely if needed.
Line 221: ‘As shown in Fig.3’, put Table 3 behind Fig.3. Fig 3 could not explain the relative expression of selected genes from one group as shown in Fig 3, while Table 3 has the relative expression. Or provide relative expression of genes in two groups, not one.
Line 310: Please check the format of all references carefully.
Figure 2 legend:arethe, please check the grammar.
Finally, professional English speaker is needed to polish the manuscript.

Experimental design

No.

Validity of the findings

No.

Additional comments

The manuscript describes a meaningful study for investigating proteins related to milk fat synthesis, which might be useful and important in dairy production. However, mistakes which needed to be clarified or corrected in this manuscript mean authors should take it seriously.

Reviewer 2 ·

Basic reporting

no comment

Experimental design

no comment

Validity of the findings

no comment

Additional comments

The authors compared the proteome in BMECs before and after knocking down VPS28, and found 203 proteins were screened as significantly down- (111) and up- (92) regulated in VPS28 knockdown BMECs, which were enrich in the “proteasome”, “ubiquitylation”, “metabolism of fatty acids”, “phosphorylation”, and “ribosome”. A series of genes associated with milk fat synthesis, ubiquitylation and proteasome pathways were analyzed by qRT-PCR. The results were interesting and provided information of the role of VPS28-associated ubiquitylation in milk fat synthesis, however, there are some issues that need to be revised.
Major comments:
1) Why only by 26% was VPS28 expression in BMEC down-regulated with tandem constructs? Generally knockdown efficiency should reach 80%.
2) Figure3, provide the SEM of the mean.
Minor comments:
1) L91, change “BEMCs” to “BMECs”
2) L169, delete “and”
3) L237, change “VPS28 was a subunit of ESCRTs played a” to “VPS28 was a subunit of ESCRTs that played a”
4) L259, after what treatment was ubiquitinated CD36 level increased significantly?
5) L278, are fatty acids activated to produce palmitate? What do you mean “fatty acids are activated”?
6) L325, 327, and 330, supply the information of doi.

---

## Round 0.2 · accepted · Accept

Dear Dr. Liu,

Thank you for your submission to PeerJ. I am pleased to inform you that your manuscript has been accepted for publication.